# Influence of Forest Conditions on the Spread of Scots Pine Blister Rust and Red Ring Rot in the Priangarye Pine Stands

**Andrey I. Tatarintsev, Pavel I. Aminev, Pavel V. Mikhaylov *** 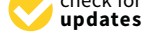 **and Andrey A. Goroshko**

Scientific Laboratory of Forest Health, Reshetnev Siberian State University of Science and Technology, 31, Krasnoyarskii Rabochii Prospekt, 660037 Krasnoyarsk, Russia; tatarintsevai@sibsau.ru (A.I.T.); aminevpi@sibsau.ru (P.I.A.); goroshkoaa@sibsau.ru (A.A.G.)
* Correspondence: mihaylov.p.v@yandex.ru

**Abstract:** Scots pine blister rust and red ring rot are common on Scots pine throughout its entire range. Specialists do not explain a significant variation in the prevalence of the diseases uniquely since it depends on complex ecological and silvicultural factors. The aim of this research is to study the influence of forest growth conditions on the incidence of Scots pine blister rust and red ring rot in pine stands of the Priangarye (territory located along the lower reaches of the Angara within the Krasnoyarsk Krai). The research methods included a detailed forest pathological examination of prevailing pine forest types, specific symptom-based macroscopic diagnosis of the diseases, data analysis using parametric and non-parametric statistical tests. Forest growth conditions indicators included type of forest, habitat conditions, and bonitet class of forest stands. The incidence of Scots pine blister rust and red ring rot in pine forests of the Priangarye reaches the extent of moderate and severe damage, respectively. The prevalence of Scots pine blister rust is significantly higher in low-bonitet lichen pine forests; the incidence rate increases along the gradient of decreasing fertility and soil moisture level. The incidence of red ring rot is significantly higher in herb-rich pine forests, in gradations of maximum soil fertility and medium soil moisture. The revealed patterns are explained by the bioecological characteristic features of pathogens (for red ring rot—additionally by factors of structural immunity in pine trees). The results of the research should be recognized in the organization of forestry practice.

**Keywords:** pine forest; Scots pine blister rust; red ring rot; forest pathological examination; incidence; forest type; type of habitat; edaphic conditions; bonitet classes



## 1. Introduction

Scots pine (*Pinus sylvestris* L.) is adapted to a wide variety of soils and climatic parameters and is one of the world's most widely distributed conifer species. It occupies a range from the arid mountain regions of Spain and Turkey to the subarctic forests of Northern Scandinavia and Siberia [1]. Priangarye pine stands are the most well-known pine forests in Siberia. A significant part of Priangarye pine forests is located within the Krasnoyarsk Krai. So-called Krasnoyarsk Priangarye is the territory along the lower reaches of the Angara River within the Krasnoyarsk Krai. Long-term forestry exploitation and other anthropogenic impacts on the Priangarye forests led to the transformation of pine stands [2–4] but did not change the tendency towards the prevalence of Scots pine in this part of the Central Siberian region [5]. Along with other conifers, mature and overmature pine forests take 60% of all pine stands in the region [5].

The state and resource potential of *P. sylvestris* stands largely depend on dendrophilic insects and dendropathogenic organisms, affecting the structure, biological diversity, and forest ecosystem functioning [6–10]. Among the most well-known diseases in pine stands are Scots pine blister rust (pathogens—micromycetes of the *Cronartium* genus) and red ring rot (pathogen—*Porodaedalea pini* (Brot.) Murrill), which are being monitored and studied in the forests of many countries in the northern hemisphere [11–19], including

Russia [20–26]. For instance, Scots pine blister rust is ranked among the leading causes of forest damage in northern Finland and Sweden [14,19,27–32]. Scots pine blister rust and red ring rot are the dominant diseases in the Priangarye pine stands [33,34]. They weaken trees, cause pathological litterfall accumulation (snag, rotten windbreak) and timber assortment reduction in commercial pine forests. Loss of timber caused by red ring rot in overmature pine forests of the Priangarye is on average 42 m$^3$/ha [35], and the loss in log volume during bucking of rotten trees is 40–75% [36].

Environmental factors, including forest growth conditions, influence these diseases' distribution and development, which explains a significant variation in the prevalence of Scots pine blister rust and red ring rot in pine forests of particular regions [16,37–40]. Some researchers indicate an increased prevalence of Scots pine blister rust [16] and red ring rot [37,39,41] in forests growing on fertile soils with an optimal moisture regime. The incidence rate in pine forests decreases with a deterioration in site conditions (dry or waterlogged low-fertile soils). Some studies reveal the most significant incidence rate of Scots pine blister rust [22,39] and red ring rot [42] in pine forests growing in dry conditions. According to other authors [43–45], the incidence of red ring rot increases when moving from more dry forest types to more humid ones. Zhukov [46] noted that the incidence rate of red ring rot in pine stands depends little on the specific types of forest, though rot foci are more common in high humidity sites. In Scotland, pine forests growing on nutrient-rich dry soils are most affected by Scots pine blister rust [47]. Some researchers [20,38,48] point to the absence of a reliable dependence of pine forests' infestation by Scots pine blister rust and red ring rot on the site conditions.

Considering the inconsistency of the available information on this issue and its ecology and forestry significance, the purpose of our research is to study the influence of site conditions on the prevalence of Scots pine blister rust and red ring rot in pine forests of the Priangarye.

## 2. Materials and Methods

### 2.1. Study Area and Objects

The research was carried out in the southern part of Krasnoyarsk Priangarye on the territory of four forest fund holders: Usolskoe, Dzerzhinskoe, Abanskoe, and Nevonskoe (Figure 1).

The object of research was pine forests of the predominant forest types: lichen (including forest community types: Bearberry/Lichen, Lingonberry/Bearberry, Lingonberry/Lichen); mossy (Moss, Lingonberry/Moss, Blueberry/Moss); herb-rich (Moss/Herbaceous, Herbaceous, Lingonberry/Herbaceous, Sedge/Herbaceous, Blueberry/Herbaceous). According to Korotkov's forest community zoning scheme [49], the studied forest stands belong to the southern taiga and subtaiga light coniferous forests of the Priangarye District of the Angara-Tunguska forest province of taiga and the forest-steppe of the Kansk-Krasnoyarsk-Biryusinsky forest province. The research covered stands of different ages (middle-aged—over-mature), density (0.3–0.9), and bonitet (I–IV), differing in site conditions (trophic state and soil moisture levels). Low density (less than 0.5) in some pine stands is due to earlier conducted selective felling. The study area is represented by pure pine forests and mixed pine forests with an admixture (up to 5 units in total) of larch, birch, aspen, and spruce. Commercial stands dominate in the region; pine forests belonging to different categories of protective forests are less represented.

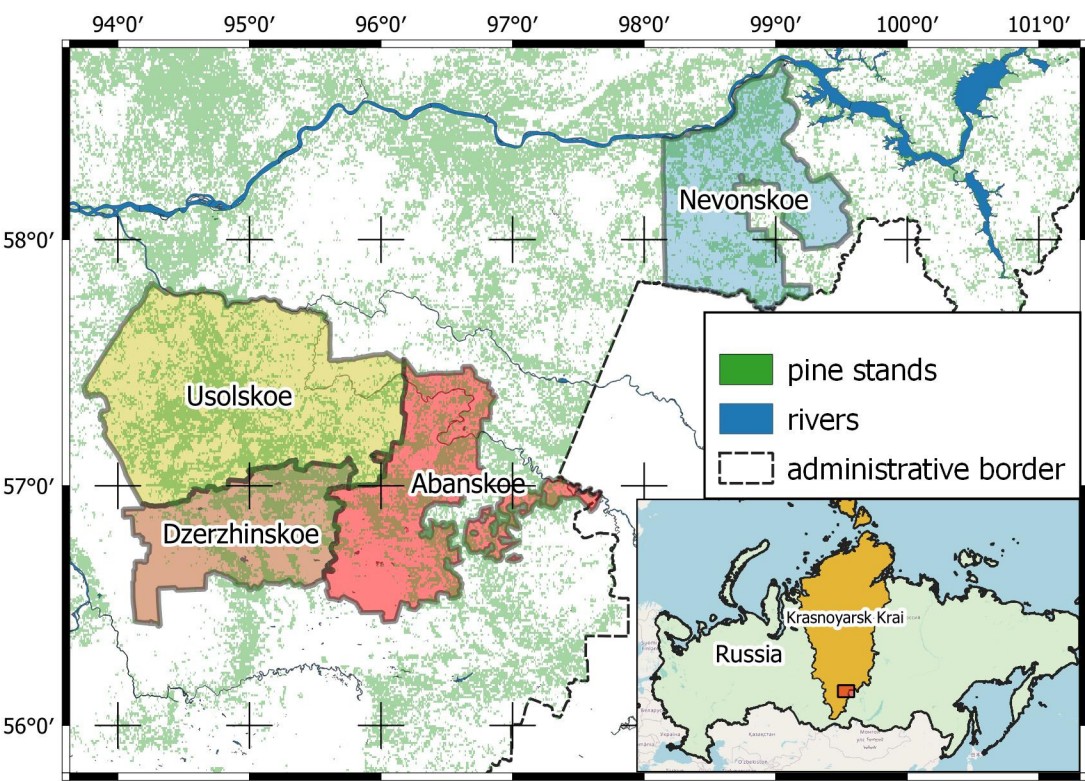

**Figure 1.** Study area.

### 2.2. Fieldwork Methods

The data were obtained during field research, which included route (reconnaissance) and detailed forest pathological surveys of the forest stands. We conducted the detailed survey following generally accepted methods [50–52] on sample plots. We used the rule of layer-by-layer sampling when placing sample plots and combined random and systematic methods of their establishment. Such techniques made it possible to cover the most distinctive forest areas in terms of landscape and site conditions within the studied area, meaning forest stands differing in inventory indicators and other parameters. We used two types of sample plots: rectangular and dimensionless, placed along the route. We included in each sample plot at least 200 stems of the main forest-forming tree species. In case the proportion of affected trees or litterfall was more than 10%, we reduced the number of trees on a sample plot to 100–150. Table 1 demonstrates the number of sample plots established in the studied pine forests and reports the disease.

**Table 1.** The number of plots (75 in total): 63 infected with Scots pine blister rust and 75 affected by red ring rot.

| Studied Diseases | Total | Including Distribution by Pine Forest Types | | |
|---|---|---|---|---|
| | | Lichen | Mossy | Herb-Rich |
| Blister rust | 63 | 13 | 24 | 26 |
| Red ring rot | 75 | 22 | 27 | 26 |

We carried out the detailed silvicultural and forest inventory description of forest vegetation on the sample plots. Forest survey was conducted by a continuous enumeration of trees by four-centimeter thickness steps and indicating condition classes. The condition class represents an integral evaluation of the state of trees by visual assessment (crown density and color, the presence and proportion of dry branches, the state of bark, etc.):

1—no signs of weakening; 2—weakened; 3—severely weakened; 4—drying up; 5—snag, including the current year and previous years. We also recorded the incidence rate when counting.

Tree diseases were diagnosed using reference literature and field guides [53–55] according to a set of direct symptoms. Symptoms of Scots pine blister rust include dark wounds with resin stains on stem and branches in different parts of the crown. Girdling of the stem may result in the death of the top or the entire tree. The rust is primarily spread by windborne spores in the form of blister-like pustules, containing orange-yellow spores usually along the periphery of the wounds. Symptoms of red ring rot in pine include swollen knots produced by *P. pini*. Dormant stem rot (lack of obvious signs) in overmature pine stands was indicated by a set of secondary signs revealed by Zhuravlev [56]: longitudinal cracks on old-looking bark, a large number of rotten branches along a considerable length of stem, or a gently curved trunk. In other cases, when determining the actual infestation of pine forests with red ring rot, a correction for dormant rot was introduced by analyzing the state of heartwood in core samples taken with an auger at the height of 1.3 m from 10% of trees from each stage of thickness, in proportion to their representation on a sample plot.

*2.3. Materials Analysis*

The incidence (damage to forest stands) was determined as the proportion (%) of affected trees (pieces) from the total sample.

The processing and analysis of the obtained data were carried out using statistical methods and criteria. We used the Kolmogorov-Smirnov test ($d_{K-S}$) to check if samples followed the normal distribution. Parametric tests were used when processing samples that resembled a normal distribution. In particular, Pearson's correlation coefficient (*r*) was used to measure the statistical relationship. We used nonparametric tests for small-size samples ($n < 10$) or samples that did not fit a normal distribution. Namely, a comparative analysis was conducted using the Mann-Whitney U test; the Kendall rank correlation coefficient was used for the statistical dependence test. In the present study, we referred to statistically significant as $p \leq 0.05$. Calculations were performed using Microsoft Excel 2010 and STATISTICA 10 programs.

**3. Results**

As a result of the conducted surveys, we established widespread but unequal distribution of Scots pine blister rust and red ring rot in pine forests of the Krasnoyarsk Priangarye. Table 2 confirms the earlier assertion, giving data differentiated by pine forest types. However, the data in Table 2 do not allow us to objectively judge the differences in the incidence of red ring rot in the Priangarye pine forests of different forest community types. This is due to the disproportionate representation within the indicated groups of the data on stands of different ages, which is essential for red ring rot incidence in insignificant wood decay. In this regard, we calculated weighted average age (age groups) values of pine stands affected by red ring rot (Figure 2).

**Table 2.** The incidence in pine forests, % (numerator—mean ± standard deviation; denominator—extreme values).

| Studied Diseases | Including Distribution by Pine Forest Types | | |
|---|---|---|---|
| | Lichen | Mossy | Herb-Rich |
| Blister rust | 11.4 ± 2.0 <br> 1.9–28.2 | 4.2 ± 0.6 <br> 0.4–12.7 | 4.8 ± 0.9 <br> 0–16.5 |
| Red ring rot | 31.1 ± 3.3 <br> 0–51.4 | 37.6 ± 3.4 <br> 4.4–70.0 | 44.2 ± 3.6 <br> 14.9–75.0 |

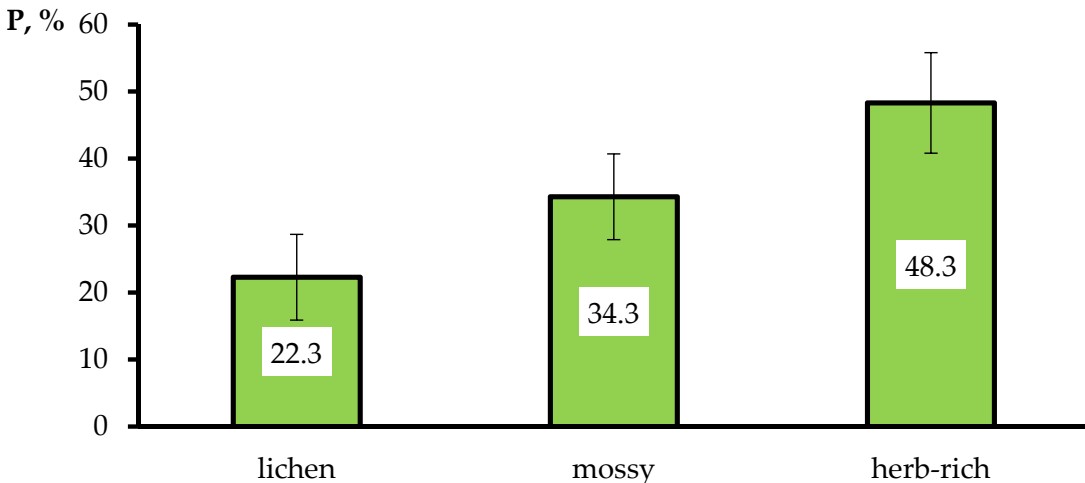

**Figure 2.** Weighted average (by age) prevalence (P) of red ring rot in different forest types. Data are the mean values ± standard error.

We assessed the reliability of differences in the prevalence of diseases by forest typological groups using the Mann–Whitney U test (Table 3) considering the statistical criteria of the analyzed samples, namely: for Scots pine blister—non-normal distribution of the sample for the herb-rich type ($d_{K-S}$ = 0.239 ($p < 0.05$)), for red ring rot—small sample size of weighted average values ($n = 6$).

**Table 3.** Disease incidence assessment in pine forests (the Mann–Whitney U test ($p$-value)).

| Compared Samples Distributed by Forest Types | Studied Diseases | |
|---|---|---|
| | Blister Rust | Red Ring Rot |
| lichen—mossy | 46 (<0.05) | 10 (>0.05) |
| lichen—herb-rich | 63 (<0.05) | 5 (<0.05) |
| mossy—herb-rich | 290 (>0.05) | 11 (>0.05) |

To characterize edaphic conditions (edaphotop), we used Vorobjev-Pogrebnyak's indicator tables (edaphic grid), showing that types of habitat conditions (soil conditions) are characterized by gradations of trophogenic (soil fertility) and hydrogenic (soil moistening) rows. Types of forest conditions for the studied pine stands fit into soil fertility gradations with an increase in fertility: pine barrens (A), pine forests growing on relatively nutrient-poor soils (B), multi-storied mixed (coniferous/deciduous) forests dominated by *P. sylvestris* L. growing on relatively nutrient-rich soils (C), according to soil moisture content—dry (1), slightly moist (2), moist (3). Figure 3 shows the mean values of the studied diseases' prevalence, corresponding to edaphic conditions indicators' gradations. The diagrams indicate differences in the incidence in pine stands depending on the type of growing conditions. We assessed the reliability of the differences by the Mann–Whitney U test (Table 4), based on the compared samples' features mentioned above.

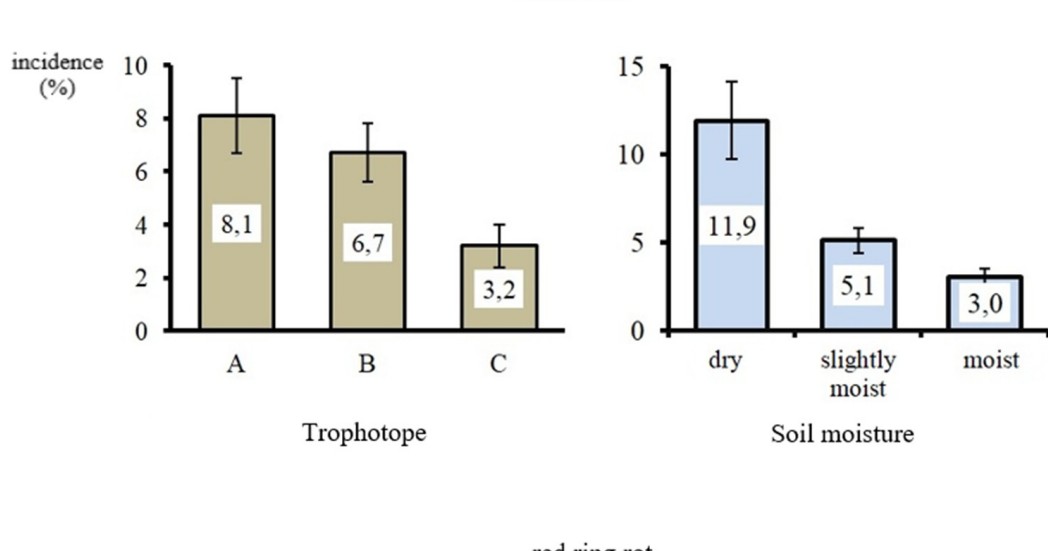

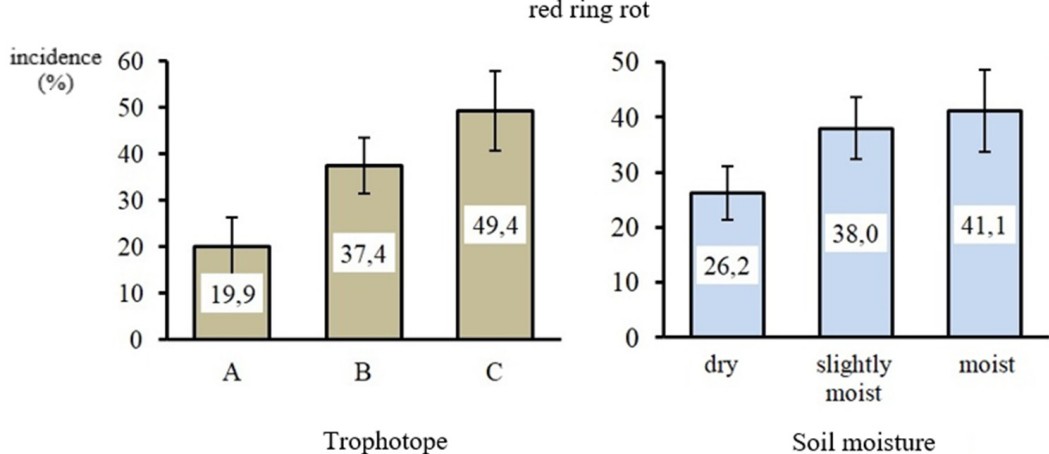

**Figure 3.** Disease infection rate in surveyed pine forests under different soil fertility (trophotope) and soil moisture. A: pine barrens, B: pine forests growing on relatively nutrient-poor soils, C: mixed forests growing on relatively nutrient-rich soils. Data are the mean values ± standard error.

**Table 4.** Assessment of differences in the disease infection rate in surveyed pine forests under different gradations of edaphotope (the Mann–Whitney U test (*p*-value)). A: pine barrens, B: pine forests growing on relatively nutrient-poor soils, C: mixed forests growing on relatively nutrient-rich soils.

| Compared Samples in the Gradations of Edaphotope | Studied Diseases | |
|---|---|---|
| | **Blister Rust** | **Red Ring Rot** |
| | trophotope | |
| A–B | 130 (>0.05) | 8 (>0.05) |
| A–C | 29 (<0.05) | 4 (<0.05) |
| B–C | 171 (<0.05) | 10 (>0.05) |
| | soil moistening | |
| dry—slightly moist | 86 (<0.05) | 2 (<0.05) |
| dry—moist | 14 (<0.05) | 4 (>0.05) |
| slightly moist—moist | 180 (>0.05) | 11 (>0.05) |

Site conditions, in particular edaphic, determine the productivity of forest stands, the forest mensuration index of which is the bonitet class. Thus, the bonitet class indirectly reflects the quality of particular site conditions through the known metric parameters: average age and height. We analyzed the entire initial material to reveal the correlation

between the prevalence of the studied diseases and the bonitet class of pine stands (Table 5). Before choosing the appropriate correlation coefficients, we checked if samples were normally distributed. We indicated a significant correlation between the bonitet class and the occurrence of Scots pine blister rust.

**Table 5.** Correlation between incidence and bonitet class.

| Studied Diseases | Checking Samples (Incidence) for Normal Distribution: $d_{K\text{-}S}$ (*p*-Value) | Correlation Coefficient (*p*-Value) |
|:---:|:---:|:---:|
| Blister rust | $d_{K\text{-}S} = 0.195$ ($p < 0.05$) | 0.345 ($p < 0.05$) [1] |
| Red ring rot | $d_{K\text{-}S} = 0.047$ ($p > 0.05$) | −0.047 ($p > 0.05$) [2] |

[1] Kendall's coefficient; [2] Pearson correlation coefficient.

## 4. Discussion

The studied diseases affected forests of the Krasnoyarsk Priangarye to a different extent: from few diseased trees (we detected the complete absence of infested trees within the sampling set only in few plots) to the formation of foci (Table 2): Scots pine blister rust—moderate damage (21–30%); red ring rot—severe damage (more than 30%). The observed variation in the prevalence of the studied diseases in the Priangarye pine forests is consistent with the information for pine stands in other regions. Such a variation is due to the differences in site conditions, forest inventory parameters, other indicators, and environmental factors.

Forest growth conditions determine plant community in all its components, including stand composition and structure (forest stand—is a major forest ecosystem engineer), and field layer. According to Sukachev's classification, forest stand composition and field layer identify forest type. In this regard, the type of forest is, to a certain extent, an indicator of the corresponding site conditions. Elucidation of the dynamics of incidence in pine forests, considering forest community types, is of scientific and practical interest. According to Sukachev [57], "the homogeneity of both the properties of the components of biogeocenoses and the properties of biogeocenosis as a whole, combined into one type, requires, under the same economic conditions, the use of homogeneous forestry activities".

Table 2 shows that in the Krasnoyarsk Priangarye, lichen pine forests were noticeably more affected by Scots pine blister rust, growing on dry low-humus sandy (sandy loam) soils. Some authors from other regions also confirm this pattern in their studies [22,39]. The incidence rate in lichen pine forests reached the foci state in half of the cases, including damage over 20%. Mossy and herb-rich forest types, usually confined to the most optimal conditions for pine, were, on average, equally affected; at the same time, the proportion of trees affected by Scots pine blister rust rarely exceeded 10%. The reliability of differences in the infestation of stands of different forest types with Scots pine blister rust was confirmed by the Mann–Whitney U test (Table 3).

Weighted average age values indicated an increase in the incidence rate of red ring rot in the forest type sequence shown in Figure 2, which had a different trend than the incidence of Scots pine blister rust. The prevalence of red ring rot is significantly higher in herb-rich pine forests than in lichen ones (Table 3).

Despite the apparent interconnection of the concepts "type of forest" and "type of site conditions", it is incorrect to identify them completely. Within the same type of site conditions, there can be several types of forests; at the same time, each type of forest may have its unique climatic conditions and soil types [58]. The considered groups of pine forest types in the Priangarye region (each of which includes several closely related forest types) correspond to a specific mosaic of site conditions, primarily edaphic.

Figure 2 shows that within the indicated gradations of the edaphotop, there is a trend towards a decrease in the infestation of pine forests with *Cronartium flaccidum* (Alb. & Schwein.) G. Winter (formally *C. pini* (Willd.) Jørst) as the fertility and soil moisture

increase. At the same time, a comparative analysis of studied samples showed significant differences in the incidence between pine forests growing on the most fertile soils compared with other gradations of the trophotop; on dry soils, slightly moist and moist (Table 4). When considering the data in Figure 2 and Table 4, we revealed a significantly higher incidence of red ring rot in pine forests corresponding to the gradations of maximum soil fertility and medium moisture (in comparison with the edaphotop of minimum fertility and low moisture). Other researchers noted a similar pattern [37,39,41]. Thus, in the Priangarye, pine forests of $A_1$–$B_1$ and $C_2$ types of site conditions are characterized by an increased prevalence of Scots pine blister rust and red ring rot.

Revealed patterns in the incidence of pine stands depending on site conditions are likely to be associated with the characteristic features of pathogens and host plants and the specifics of their interactions. In the Krasnoyarsk Priangarye, Scots pine grows in optimal conditions despite certain heterogeneity, so stands productivity is relatively high in the region. Rust fungi (causing Scots pine blister rust) are obligate parasites, meaning that they do not need to weaken plants since they need a living host plant for development and are capable of infecting any tree in a stand, including perfectly healthy ones [38–40,59]. In contrast, the weakening of host plants is essential for the onset of the pathological process for hemiparasitic pathogens causing necrosis and desiccation of tree tissue (facultative saprophytes and parasites) [60]. On dry, marginal soils, pine stands having a species-poor field layer (lichen type of forest) are formed where grow no possible intermediate hosts for *C. flaccidum*—plants of the genera *Vincetoxicum*, *Pedicularis*, *Impatiens*, *Verbena*, and others [16]. Only *C. pini* (rust fungi that produce all spore forms on one species of host plant) causes Scots pine blister rust in the study area. *C. pini* aeciospores infect a pine host without the intermediate, ensuring a relatively rapid spread of infection and lesions appearing on groups of trees. Additionally, pathogens that cause Scots pine blister rust are thermophilic and light-demanding [40,59,61,62]. Under the appropriate conditions, pathogens develop more actively in the host tissues and form abundant aeciospores, accelerating the spread of disease. In this respect, pine forests growing on dry soils with a warmer microclimate create more favorable hydrothermal conditions for pathogen growth and disease spread.

We stick to the view of Romanovsky and colleagues [63] when explaining the increased prevalence of red ring rot in herb-rich pine forests, which are usually formed on fertile soils with an optimal moisture regime. Romanovsky and colleagues stated that fast-growing trees growing in the best edaphic conditions are mostly affected by red ring rot due to wide-ringed and loose-textured wood, which, in contrast to fine-grained wood, is characterized by a reduced percentage of latewood and resins that prevent wood biodegradation. At the same time, fast-growing trees have an increased metabolic rate, including a high redox potential. Apparently, structural immunity factors are of primary importance in pine resistance to red ring rot. Additional factors contributing to pine stands infection by red ring rot are age-related changes in the sapwood to heartwood ratio (increased heartwood volume), large dead branches, trunk wounds (blaze cut, burns, etc.), serving as gates to infection [34,64,65].

## 5. Conclusions

In the forests of Priangarye, which was primarily affected by logging in the second half of the 20th century, Scots pine stands remain predominant and economically significant, despite disturbance due to felling and forest fires. In pine forests, where more than 60% are mature and overmature stands, the primary diseases leading to the weakening and loss of trees are Scots pine blister rust and, to a lesser extent, red ring rot. The red ring rot, reaching slow epiphytotic, is the main reason for the decrease in the output of commercial Scots pine stands.

Scots pine blister rust significantly affects low-bonitet lichen pine forests. The prevalence of the disease increases along the gradient of decreasing soil fertility and soil moisture. The decrease in litterfall species richness contributes to single-host infectious agents spreading directly from tree to tree. Based on the identified ecological and silvicultural patterns,

the determining factors in the development of Scots pine blister rust are factors that directly or indirectly affect disease causative agents, specifically obligate parasites (micromycetes of the *Cronartium* genus) and the possibility of their presence in the phytocenosis, intensity on sporulation, accumulation of infection, and incidence rate.

Several conditions favor red ring rot development (the causative agent *Porodaedalea pini* is a hemiparasite that enters trees through wounds), including the state of the host plant (sapwood/heartwood ratio, the structure of wood, latewood ratio), and ways to infect the heartwood (dead branches, wounds, etc.). This explains a significant increase in the incidence of red ring rot in pine forests in the following sequence of forest types: lichen–herb-rich, as conditions for pine growth, are optimized.

The revealed patterns should be taken into account when planning silviculture and other forestry activities in pine forests to improve their sanitary condition and optimize commercial forestry.

**Author Contributions:** Formal analysis, A.I.T., P.I.A., P.V.M. and A.A.G.; investigation, A.I.T. and P.V.M.; methodology, A.I.T. and P.I.A.; writing—original draft, A.I.T.; writing—review & editing, A.I.T. and P.V.M. All authors have read and agreed to the published version of the manuscript.

**Funding:** The research was carried out within the projects "Fundamentals of forest protection from entomo- and fittings pests in Siberia" (№ FEFE-2020-0014) within the framework of the state assignment, set out by the Ministry of Education and Science of the Russian Federation, for the implementation by the Scientific Laboratory of Forest Health.

**Institutional Review Board Statement:** Not applicable.

**Informed Consent Statement:** Not applicable.

**Data Availability Statement:** The data presented in this study are available on request from the corresponding author.

**Conflicts of Interest:** The authors declare no conflict of interest. The funders had no role in the design of the study; in the collection, analyses, or interpretation of data; in the writing of the manuscript; or in the decision to publish the results.

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
