# Peer review of "Influence of Forest Conditions on the Spread of Scots Pine Blister Rust and Red Ring Rot in the Priangarye Pine Stands"

_land, doi:10.3390/land10060617_

Round 1

Reviewer 1 Report

Dear Authors,

My general impression of this study and its presentation in the form of a manuscript is positive, but some adjustments are necessary.

Arrange the manuscript following the journal's instructions. Discussion reports data from “materials and methods” and should be improved.

Some typos are present.

In general:

I suggest the term “diseases” instead of “pathologies” (lines 48, 53, 187).

The current name of Cronartium flaccidum (Alb. & Schwein.) G. Winter should be Cronartium pini (Willd.) Jørst.

Introduction

Line 38: Insert a reference.

Lines 41-43: Use “Among the most well-known pathogens in pine stands are Cronartium flaccidum (Alb. & Schwein.) G. Winter (micromycetes agent of blister rust) and Porodaedalea pini (Brot.) Murrill (associated to red ring rot)” instead of “Among … Murrill)”

Line 61: use “Authors” instead of “specialist”.

Line 61: What does "drier" means?

Lines 66-67: What does "the studied" means?

Materials and Methods

Line 74: use “Abanskoe and Nevonskoe” instead of “Abanskoe, Nevonskoe”

Lines 74-75: delete “forestries”.

Results

This section reports data from “materials and methods”. In addition, all the section is very confused.

Table2: See table 1 comments. I suggest reporting data as mean ± standard deviation or as minimum (mean) and maximum.

Line 162: What does "average statistical values" means?

Figure 3: Use “blister rust” instead of “1” and “red ring rot” instead of “2”. Use “incidence (%)” instead of “P, %”. Delete “A, B, C,

Figure 3 legend: Disease infection rate in surveyed pine forests under different soil fertility (habitat topotype) and soil moisture. A: pine barrens, B: pine forests growing on relatively nutrient-poor soils, C: mixed forests growing on relatively nutrient-rich soils.

Table 4: What does "A, B, C" means?

Table 4: This table is present on two pages.

Table 5: What do "* and "**" mean?

Discussion

Line 192: delete "V.N." and insert reference number [57].

Line 196: delete "V.N."

Line 219: Use “C” instead of “Cronartium”.

Line 240: delete " fungus" and insert a reference.

Lines 302-311: Delete.

Arrange the references following the journal's instructions.

Best regards

Author Response

The authors are grateful for a careful reading of the work and the comments made. We really appreciate the informational help that will help improve our manuscript.

I suggest the term “diseases” instead of “pathologies” (lines 48, 53, 187).

The mentioned lines have been changed:

49-50 Scots pine blister rust and red ring rot are the dominant diseases in the Priangarye pine stands [33, 34].

55: Environmental factors, including forest growth conditions, influence these diseases' distribution and development…

209: The observed variation in the prevalence of the studied diseases in the Priangarye pine forests…

The current name of Cronartium flaccidum (Alb. & Schwein.) G. Winter should be Cronartium pini (Willd.) Jørst.

The respected reviewer is right - the current species name of the fungus is Cronartium pini (Willd.) Jørst. However, we consider it appropriate to indicate a fungus by a synonym Cronartium flaccidum considering differences in their life-cycles (Alb. & Schwein.) G. Winter.

Line 38: Insert a reference.

The references was inserted.

Lines 41-43: Use “Among the most well-known pathogens in pine stands are Cronartium flaccidum (Alb. & Schwein.) G. Winter (micromycetes agent of blister rust) and Porodaedalea pini (Brot.) Murrill (associated to red ring rot)” instead of “Among … Murrill)”

In the mentioned sentence and paragraph, the emphasis is on diseases rather than pathogens. Based on your remarks, this sentence has been edited:

Among the most well-known diseases in pine stands are Scots pine blister rust (pathogens - micromycetes of the genus) and red ring rot (pathogen - Porodaedalea pini (Brot.) Murrill), which are being monitored and studied in the forests of many countries in the northern hemisphere [11, 12, 13, 14, 15, 16, 17, 18, 19], including Russia [20, 21, 22, 23, 24, 25, 26].

Line 61: use “Authors” instead of “specialist”.

The sentence has been changed. See the line 63.

Line 61: What does "drier" means?

We are talking about dry types of forest, in other words - arid forest conditions. Probably, we used comparative adjective incorrectly. The sentence has been changed:

…the incidence of red ring rot increases when moving from more dry forest types to more humid ones

Lines 66-67: What does "the studied" means?

See the line 67-69

Some researchers [48, 38, 20] point to the absence of a reliable dependence of pine forests' infestation by Scots pine blister rust  and red ring rot on the site conditions.

Line 74: use “Abanskoe and Nevonskoe” instead of “Abanskoe, Nevonskoe”.

Lines 74-75: delete “forestries”.

The sentence has been changed:

The research was carried out in the southern part of Krasnoyarsk Priangarye on the territory of four forest fund holders: Usolskoe, Dzerzhinskoe, Abanskoe and Nevonskoe (Figure 1).

This section reports data from “materials and methods”. In addition, all the section is very confused.

Some methodological extracts in the Results section were placed intentionally to reflect the conceptual logic in presenting the main results.

The information is presented in this section as follows: data on the incidence of diseases in prevailing pine forest types; statistical analysis of differences in the prevalence of diseases by forest types → data on the prevalence of diseases by types of forest growth conditions; analysis of the reliability of differences in the prevalence of diseases according to the gradations of edaphic conditions → analysis of the relationship between the incidence of diseases in pine forests and the bonitet of forest stands.

Table2: See table 1 comments. I suggest reporting data as mean ± standard deviation or as minimum (mean) and maximum.

We changed comments to Table 2.

Line 162: What does "average statistical values" means?

We edited id as mean values.

Figure 3: Use “blister rust” instead of “1” and “red ring rot” instead of “2”. Use “incidence (%)” instead of “P, %”. Delete “A, B, C.

Figure 3 and its legend have been changed according to the comments made.

Table 4: What does "A, B, C" means?

Table 4: This table is present on two pages.

Table 4 and its legend have been changed according to the comments made.

Table 5: What do "* and "**" mean?

Table 5 and its legend have been changed.

Line 192: delete "V.N." and insert reference number [57].

Line 196: delete "V.N."

The initials were excluded from the manuscript.

Line 219: Use “C” instead of “Cronartium”.

See line 243.

Line 240: delete " fungus" and insert a reference.

See line 264

Lines 302-311: Delete.

The mentioned lines were delited.

Reviewer 2 Report

The manuscript titled Influence of forest conditions on the spread of Scots pine blis ter rust and red ring rot in the Priangarye pine stands“ by Tatarintsev and co-authors presents study aimed to study the influence of forest conditions on the incidence of Scots pine blister rust and red ring rot in pine stands of the Priangarye. I find the work interesting and in line with the aim of the journal. The abstract is poorly written, it should contain an introduction aim hypothesis aim result, and conclusion. The introduction section is too long in the abstract; one line of the background of study in the abstract attracts the reader most. A connective link is missing between different sections. Also, the concluding part of the introduction is missing at the end of the introduction. The author should make the introduction section crisp and to the point related to research, which I don't find in the present form of the manuscript.  I have some concerns on the experimental setup to justify what the authors claim. Moreover, the rationale behind some of the data presented was not entirely clear. I also recommend to the authors to improve their references by conducting a more extensive review on international literature.  Below are my point-to-point analysis of the manuscript.

  • I suggest modifying the title should be more crisp and brief.

  • Abstract introductory statement is too long, it has to be improved with a more specific rationale of the study. The abstract should have crisp information about the aim materials method result and conclusion, which I don't find.
  • Statment „ Major pine forests in Siberia are located in the Kras noyarsk Priangarye – territory along the lower reaches of the Angara within the Krasno yarsk Krai“ needs a citation
  • Statment „Scots pine blister rust and red ring rot are the domi48 nant pathologies in the Priangarye pine stands“ need to be cited
  • Conclusion in Introduction is missing.
  • Statics is combined with material and method which is confusing to the reader it should be in a seprate heading.
  •  

  • My main concern of a manuscript is the statical test.

what is the value of n while calculating ANOVA?

n Value (3) used in the manuscript is too few to examine the normal distribution of variables in the sample, however, Shapiro-Wilk test is appropriate for samples from 3 to 5000 but for lesser value of n it receives non-normal distribution. Thus ANOVA that is a parametrical test is incorrect for such small samples.

  • The author should mention the data set that does not pass the normality test.
  • The author has applied parametric and nonparametric test, he has not explained the reason for non-parametric tests is it because of nonnormal distribution or because of variance.

  • I have doubt with the stat done as the statistic is the backbone of any research study.

  • Secondly, the error bar in the figure No 2 and 3 are not explained it seems it correspond to the SD which does not make any sense, that is merely for decorative purpose. I highly recommend using confidence interval insted of Standard deviation, in the error bar.

  • Although the study is interesting and could be useful for a certain group of scientific fertility, therefore, I would suggest improving the manuscript substantially, giving a chance for the next round, because the subject is interesting. However, even an interesting subject depends on statics.

Author Response

The authors are grateful for a careful reading of the work and the comments made. We really appreciate the informational help that will help improve our manuscript.

I suggest modifying the title should be more crisp and brief.

We appreciate your opinion but believe that all words in the title play an important role and would like to remain the title unchanged.

Abstract introductory statement is too long, it has to be improved with a more specific rationale of the study. The abstract should have crisp information about the aim materials method result and conclusion, which I don't find.

The abstract has been edited.

Statment „ Major pine forests in Siberia are located in the Kras noyarsk Priangarye – territory along the lower reaches of the Angara within the Krasno yarsk Krai“ needs a citation.

The mentioned sentence has been changed.

Statment „Scots pine blister rust and red ring rot are the domi48 nant pathologies in the Priangarye pine stands“ need to be cited.

We included a reference.

Statics is combined with material and method which is confusing to the reader it should be in a seprate heading.

The Materials and Methods section was divided into three parts with separate headings.

My main concern of a manuscript is the statical test.

what is the value of n while calculating ANOVA?

n Value (3) used in the manuscript is too few to examine the normal distribution of variables in the sample, however, Shapiro-Wilk test is appropriate for samples from 3 to 5000 but for lesser value of n it receives non-normal distribution. Thus ANOVA that is a parametrical test is incorrect for such small samples.

The author should mention the data set that does not pass the normality test.

The author has applied parametric and nonparametric test, he has not explained the reason for non-parametric tests is it because of nonnormal distribution or because of variance.

I have doubt with the stat done as the statistic is the backbone of any research study

We used the Kolmogorov-Smirnov test (dK-S) to check if samples followed the normal distribution. Parametric tests were used when processing samples that resembled a normal distribution. We used nonparametric tests for small-size samples (n <10) or samples that did not fit a normal distribution. We added this information in the Materials and methods section.

We explained the usage of each test in the Results section:

We assessed the reliability of differences in the prevalence of diseases by forest typo-logical groups using the Mann–Whitney U test (Table 3) considering the statistical criteria of the analyzed samples, namely: for Scots pine blister rust – non-normal distribution of the sample for the herb-rich type (dK-S =0,239 (p<0,05)), for red ring rot – small sample size of weighted average values (n = 6).

We analyzed the entire initial material to reveal the correlation between the prevalence of the studied diseases and the bonitet class of pine stands (Table 5).

We did not use the ANOVA parametric test because compared samples did not fit into the normal distribution or were small (n <10).

Secondly, the error bar in the figure No 2 and 3 are not explained it seems it correspond to the SD which does not make any sense, that is merely for decorative purpose. I highly recommend using confidence interval insted of Standard deviation, in the error bar.

We changed the mentioned figures legends.

Reviewer 3 Report

see add. file

Author Response

The authors are grateful for a careful reading of the work and the comments made. We really appreciate the informational help that will help improve our manuscript.

What results do the authors consider as "new"?

For the first time, we used the forest inventory characteristics of the infection of pine forests with the studied diseases in the Priangarye pine forests (within the Krasnoyarsk Krai) and combined it with a large volume of original field material. All data were analyzed using modern statistical tools, including parametric and nonparametric tests. We revealed patterns of diseases infection rates depending on the forest growth conditions, although there are conflicting data on this issue in the literature. The interpretation of the obtained results is presented, taking into account the biological and ecological characteristics of pathogens.

Can the authors compare their "new" results with the data already available for Siberia, Russia and other countries? Is there a difference or not?

Both similarities and differences with the results published in the literature have been established. The consistency of our results with the data of other authors was indicated in the Discussion section.

This work is devoted to the study of the ecological structure of forests, soils, etc., is it possible to assess the similarities and differences at this level?

The present study was not devoted to the study of the ecological structure of forests, soils, etc.

What recommendations can the authors give to forest services faced with similar pathogens in Europe, the European part of Russia, Yakutia?

The aim of the present research was not to develop and propose specific recommendations for forestry. However, the results obtained can help optimize forest management when planning and carrying out forest protection measures in pine forests (primarily in the forests of Eastern Siberia). We indicate this at the end of the Conclusion section.

Reviewer 4 Report

In general, the manuscript is well prepared and well presented. 

However, there are just some minor points to consider:

Line 42 delete microcetes of the genus Cronatium because the scientific binomial has already been cited.

Line 250, 253 rephrase as V.R. Romavsky et al. Is there any reason why the initials are included. Usually the surname will do unless there is another researcher bears the same surname. This applies to the rest of the manuscript.

Author Response

The authors are grateful for a careful reading of the work and the comments made. We really appreciate the informational help that will help improve our manuscript.

Line 42 delete microcetes of the genus Cronatium because the scientific binomial has already been cited.

The mentioned line has been changed.

Line 250, 253 rephrase as V.R. Romavsky et al. Is there any reason why the initials are included. Usually the surname will do unless there is another researcher bears the same surname. This applies to the rest of the manuscript.

The initials were excluded from the manuscript.

Round 2

Reviewer 1 Report

Dear Authors,

I suggest some minor adjustment:

Line 27: use "bonitet classes" instead of "bonitet"

Line 108: Verify the title: Table reports 138 plots: 63 with blister rust and 75 showing red ring rot.

Lines 159-160: use ". Data are the mean values ± standard error" instead of ": mean value; error bars – standard error"

Line 183: use ". Data are the mean values ± standard error" instead of ": mean value; error bars – standard error"

Line 243: use Cronartium flaccidum (Alb. & Schwein.) G. Winter (formally C. pini (Willd.) Jørst) instead of " Cronartium flaccidum"

Author Response

The authors are grateful for a careful reading of the work and the comments made. We really appreciate the informational help that will help improve our manuscript.

Line 27: use "bonitet classes" instead of "bonitet".

Line 27 was edited according to the comment made.

Line 108: Verify the title: Table reports 138 plots: 63 with blister rust and 75 showing red ring rot.

Table 1 title and description were edited according to the comment made. See lines 109-110.

Lines 159-160: use ". Data are the mean values ± standard error" instead of ": mean value; error bars – standard error"

Figure 2 legend was edited according to the comment made. See lines 161-162.

Line 183: use ". Data are the mean values ± standard error" instead of ": mean value; error bars – standard error"

Figure 3 legend was edited according to the comment made. See line 186.

Line 243: use Cronartium flaccidum (Alb. & Schwein.) G. Winter (formally C. pini (Willd.) Jørst) instead of " Cronartium flaccidum"

 The line was edited according to the comment made. See lines 246-247

Reviewer 2 Report

author has revised the manuscript as per my suggestions. The manuscript doesn't need further revision I recommend accepting the manuscript in its present form. 

Author Response

Dear reviewer, the authors are grateful for the careful reading of the work and the comments made. We really appreciate the informational help that helped improve our manuscript.